# Impact of Perpetrator and Victim Gender on Perceptions of Stalking Severity

**DOI:** 10.3390/bs15020120

**Published:** 2025-01-24

**Authors:** Megan Brenik, Ana-Cristina Tuluceanu, Emma Smillie, Luan Carpes Barros Cassal, Caroline Mead, Dara Mojtahedi

**Affiliations:** 1Faculty of Health and Wellbeing, School of Psychology, University of Bolton, Bolton BL3 5AB, UK; m.brenik@bolton.ac.uk (M.B.); a.tuluceanu2@bolton.ac.uk (A.-C.T.); e.smillie@bolton.ac.uk (E.S.); c.mead@bolton.ac.uk (C.M.); 2Manchester Institute of Education, University of Manchester, Manchester M13 9PL, UK; cassal@manchester.ac.uk; 3School of Human and Heath Sciences, University of Huddersfield, Huddersfield HD1 3DH, UK

**Keywords:** stalking, victim, myths, victim blaming, sexual orientation

## Abstract

Many individuals will dismiss the seriousness of ex-partner stalking offences, often as a result of inaccurate and problematic beliefs about the offence (stalking myths). However, to date, stalking myth acceptance measurements have only considered attitudes about stereotypical stalking (male stalking a female). The current research considered whether inaccurate and problematic perceptions of stalking were dependent on the gender and sexuality of the perpetrator, victim, and participant. Additionally, it examined whether existing stalking myth acceptance scales measuring stereotypical stalking attitudes would predict perceptions of stalking incidents that involved female stalkers and/or male victims. Participants (*N* = 336) completed the stalking myth acceptance scale and then responded to a series of questions measuring their perceptions towards a stalking vignette. An independent groups design was used to manipulate the gender of the stalker and victim. The need for police intervention was greatest for incidents involving a male stalker and a female victim. Female victims of male stalking were predicted as being the most fearful, whilst male victims of female stalking were rated as least likely to be fearful. Heterosexual males and participants with minority sexual orientations were also more likely to identify the perpetrator’s actions as stalking. Finally, the SMA scales predicted participants’ attitudes for stereotypical stalking cases but not for the other scenarios. The findings demonstrate that gender plays a significant role in stalking perceptions and highlights the need for more inclusive SMA measurements to consider problematic attitudes towards non-stereotypical stalking.

## 1. Introduction

Stalking is characterised by a pattern of persistent pursuit and intrusive behaviours directed by one person towards another, which can continue for several months or even years ([31]; [30]). Though statutory definitions differ across jurisdictions, most legal definitions accept that stalking behaviours must constitute three core elements that differentiate the crime from non-illicit interpersonal behaviours (e.g., courting). These are a pattern of intrusive behaviours that is unwarranted, evidence of implicit or explicit threat, and feelings of fear by the victim as a result ([26]).

The most frequent perpetrators of stalking are ex-partners (45%, [17]), though other reports suggest a relatively similar proportion of stalkers are acquaintances or strangers ([7]). Ex-partner stalking also carries the highest risk of violence ([49]), with this stalking often perpetrated as a continuation of domestic violence and an attempt to maintain control after a relationship has ended ([49]).

Despite this, the general public are often reluctant to identify stalking behaviours that occur between former partners as stalking ([5]; [42]) and may even excuse non-violent ex-partner stalking behaviours which occur shortly after the relationship ends ([42]). This may be due to non-violent stalking behaviours (e.g., persistent attempt to communicate with former partners) being perceived as expected courting or post-break-up behaviours ([13]). Furthermore, cases of ex-partner stalking are perceived as less serious and incur greater victim blaming in comparison to cases of stranger or acquaintance stalking ([47]). Although police officers with experiences of stalking cases are less likely to engage in victim blaming, they too show greater willingness to identify stranger stalking cases as stalking compared to ex-partner stalking ([57]).

Such justifications are a result of inaccurate and biased beliefs about stalking, often referred to as *stalking myths* ([18]). As with other sexual crimes (e.g., rape myths; see [38]), many of these inaccurate beliefs are commonly held and serve to excuse stalking behaviours. For instance, reports show that the general public may place some accountability onto victims, believing that a stalker’s actions were brought on by the victim’s own decisions ([43]), or that the victim should have done more to stop the situation ([24]). The victim and offender’s pre-existing relationship is also likely to incur additional stalking myths unique to the relationship dynamic; for instance, a common stalking myth about ex-partner stalking is the belief that the recipients are somewhat culpable for the outcome due to ending the relationship and that the stalkers’ behaviours are justified ([24]; [41]).

To better understand the myths relating to ex-partner stalking, [18] ([18]) conceptualised a model of stalking myth acceptance (SMA) consisting of three different constructs. The first construct, *victim blaming*, pertains to beliefs that victims of stalking are partly responsible for their stalker’s actions (for example, believing that women who date frequently are more likely to be stalked). These views can be attributed to *just world* beliefs, a cognitive bias where an individual accepts that the consequences that befall on people are morally fitting (i.e., “bad things happen to bad people”; [23]). Research has demonstrated the existence of victim blaming in stalking perceptions, such that many individuals will assume that victims of stalking will have placed themselves into situations that resulted in their stalking ([10]; [13]; [24]; [42], [43]; [51]). Another common example of victim blaming in stalking is the fixation on sexual promiscuity as a cause of the ordeal. [10] ([10]) found that a stalking victims’ sexual history with the perpetrator was used to deflect responsibility away from the perpetrator ([10]). In the context of intimate partner stalking, victim blaming myths centre around the notion that the victims are responsible for their own stalking due to being promiscuous and/or due to ending a relationship with someone who is attached to them.

The second construct, *stalking minimisation*, reflects attitudes which serve to minimise the actual harm inflicted by stalking and downgrade such actions from being considered as a crime (for example, believing that stalking should not be labelled as a crime if there is no violence involved). Within the context of intimate partner stalking, minimising beliefs serve to downgrade stalking behaviours as attempts to rekindle a relationship. Research has shown that individuals with greater experience and training on stalking (i.e., police officers) are more likely to perceive stalking as a serious incident ([29]; [44]; [57]), suggesting that a tendency to minimise stalking behaviours could be linked to limited understanding about the offence.

The third construct, *flattery*, reflects beliefs which serve to normalise stalking as courting behaviours that are complimentary rather than problematic (e.g., the belief that women are flattered by stalking behaviours). [4] ([4]) reported that the normalisation of stalking by men against women was facilitated by culturally accepted gender narratives that present stalking behaviours as acceptable and, in some cases, flattering towards the victim.

In order to comprehensively and reliably measure stalking myths, academics have developed attitude measurement tools, such as [29]’s ([29]) stalking-related attitude questionnaire (SRAQ) and [18]’s ([18]) stalking myth acceptance (SMA) scale, which modified [29]’s ([29]) original scale items to measure ex-partner stalking attitudes more specifically. These scales have been extensively used within the literature and have contributed to furthering the understanding of stalking biases (see [11]; [56]). However, these scales only measure attitudes towards male-on-female stalking. This potentially reproduces cis-heteronormative hierarchies and discriminations.[note 1] It is unclear whether stalking myth acceptance remains static across different stalker/victim genders (i.e., are individuals who excuse stalking against women equally likely to excuse stalking against men or are those who blame female victims of stalking equally likely to place responsibility on male victims?). Research on other sexual crimes such as rape and sex trafficking indicate that victim blaming and perpetrator-excusing beliefs can stem from inherent biases towards the victim’s characteristics (e.g., sexism, [27]; [39]; or homophobia, [14]), rather than false beliefs about the crime. We understand gender as a socially constructed set but ever-changing normative framework, involving individual and collective identification and recognition ([9]). This set creates socially shared expectations about how one looks, performs, and desires ([8]); however, this framework does not explain how individual characteristics are related to how one *perceives* the world, which we try to address here. We acknowledge the usefulness of models to identify perceptions in order to challenge stereotypes. Therefore, current SMA measures may not be reliable for non-normative cases of stalking (e.g., same-gender stalking or female stalking a male[note 2]).

### 1.1. Effects of Gender and Sexual Orientation on Stalking Perceptions

Multiple studies have shown that the gender of both the victim and perpetrator can influence perceptions of severity and culpability. In a qualitative study, [24] ([24]) identified stark differences in how individuals perceived stalking scenarios, which either involved a female stalking a male (FSM) or a male stalking a female (MSF). In cases of intimate partner stalking, victim blaming was more prevalent for MSF cases and the male perpetrator’s actions were more likely to be normalised, whereas female perpetrators were seen as an unusual phenomenon and were more likely to be pathologised (e.g., behaviours attributed to mental instability). Furthermore, the seriousness of FSM cases was downplayed with male victims of stalking being seen as overreacting and not conforming to their gender expectations. Although findings from [24] ([24]) indicate that male-perpetrated intimate partner stalking is more likely to be normalised, a large body of research demonstrates that participants still perceive male-perpetrated stalking as being more dangerous than female-perpetrated stalking (e.g., [10]; [24]; [51]). Furthermore, these differences in perceptions of male perpetrators do not appear to differ between cisgender and transgender male perpetrators ([13]). This may be attributed to the criteria that many individuals and legal jurisdictions use to define stalking. As mentioned earlier, many definitions of stalking include a pre-requisite of fear and distress. Similar findings have also been observed in other intimate partner crimes such as intimate partner violence, where female-perpetrated violence is seen as comparatively less serious, which is partly due to female perpetrators being perceived as less threatening ([2]; [55]).

Though research examining the impact of sexual orientation (of perpetrators and victims of stalking) on stalking perceptions is limited, the available evidence suggests that the sexuality of the victim and perpetrator does not have any additional influence on stalking perceptions ([5]; [13]). [45] ([45]) did not find any interaction effect between victim and perpetrator gender that would suggest differences between same- and opposing-gender stalking. However, [45] ([45]) reported that stalking cases were perceived as more severe when the perpetrator was presented as a male and the victim was a female.

Perceptions of stalking incidents can also be influenced by the gender and sexual orientation of the perceiver. Research on stalking attitudes mirror studies that have examined public perceptions towards other sexual crimes (e.g., rape or ex-partner violence; [16]; [50]), whereby female participants show greater support towards victims and greater punitive attitudes towards the perpetrators when compared to their male counterparts ([45]). [45] ([45]) attributed these findings to defensive attribution theory, which posits that individuals are more likely to identify with the victim of a threatening situation if they believe that the incident could affect them in the future (see [20]). Male observers’ identification with perpetrators may be due to the endorsement of gender stereotypes, which suggest that males are required to pursue females in the formation of relationships ([29]). Research comparing stalking attitudes between different sexuality groups is comparatively scarce. However, victim reports suggest that sexual minorities are more likely to experience stalking behaviours ([12]; [14]; [19]). As a result, individuals from sexual minorities may be more inclined to empathise with victims of stalking through defensive attribution. Examining the perceptions and attitudes of individuals with minority sexual orientations[note 3] towards stalking may offer more insight into understanding how their experiences may impact their views on such behaviours.

### 1.2. The Present Study

There is continued interest in expanding the field of research to analyse perceptions of ex-partner stalking cases that deviate from “typical” male-on-female stalking. It cannot be presumed that the outcomes of some previous research, whose methodologies include the use of heterosexual vignettes, are applicable to cases involving people with minority sexual orientations. Whilst some emerging studies have examined the impact of victim/stalker gender and sexual orientation on perceptions of seriousness, the literature is limited. Thus, the first aim of the present study was to determine whether perceptions of stalking incidents (i.e., perceptions of harm, responsibility, victim fear, need for intervention, and identification as stalking) were influenced by the gender and sexual orientation of the victim and stalker. Based on previous research suggesting that male-stalking-female offences are perceived as the more dangerous (e.g., [10]; [51]), two hypotheses were made:

**Hypothesis** **1.**
*Participants will rate male-perpetrated stalking as more likely to induce fear, cause harm, and require police intervention than female-perpetrated stalking.*


**Hypothesis** **2.**
*Participants will rate stalking perpetrated against females to be more likely to induce fear, cause harm, and require police intervention than stalking perpetrated against males.*


Despite some reports suggesting that people with minority sexual orientations are more likely to experience stalking than heterosexual individuals ([28]), there is little research looking at the role sexual orientation plays in informing attitudes towards stalking. To contribute to the current dearth of research into stalking within the LBGTQ+ community, the second aim of this study examined whether the gender and sexual orientation of participants influenced their perceptions of stalking. Due to a lack of research around sexual orientation and stalking attitudes, no prediction was made. With regard to gender differences, based on defensive attribution theory, the following hypothesis was made:

**Hypothesis** **3.**
*Female participants will report attitudes that are more supportive towards stalking victims.*


Finally, although reliable models and scales to capture stalking myths surrounding male-on-female stalking have been developed, such work overlooks problematic beliefs about non-normative stalking. It is unclear whether the same individuals who hold problematic attitudes towards female victims of stalking are likely to hold the same views towards male victims. Thus, the third aim of the present study was to examine whether existing SMA scales that are based on male-on-female ex-partner stalking can also predict participant attitudes towards stalking involving non-normative dyads. There was limited research available to inform a prediction for this aim; therefore, no hypotheses were made.

## 2. Methods

### 2.1. Sample and Design

An online experiment was created using Qualtrics (Qualtrics, Provo, UT, USA) and disseminated through the internet (social media and internet forums) and through an author’s institutional participant pool platform (psychology students took part in the experiment in return for course credit). Participants did not receive any financial compensation for their involvement. An a priori power analysis G*power 3.1.9.2 ([21]) for the most sample-demanding analysis of this study (between-groups differences in stalking attitudes in experimental condition) indicated that for a medium-sized effect (*f* = 0.25) and power of 0.80, a minimum of 180 participants would be required. Initially, 343 participants took part in the experiment; however, 7 participants were removed from the dataset due to failing to answer vignette questions, leaving a usable sample of 336 participants (female = 268; male = 63; non-binary = 5) aged 18–72 (*M* = 28.62, *SD* = 11.5). Of these participants, 193 were active university students. The sexual orientation of the sample was primarily heterosexual (*n* = 280; 52 male and 228 female); the remaining participants identified as bisexual (*n* = 29; 4 male and 25 female) or homosexual (*n* = 15; 4 male, 10 female, 1 non-binary); three participants selected “other sexual orientation” (2 female and 1 non-binary) and eight participants did not disclose their sexuality (3 male, 3 female, and 2 non-binary).

Participants were required to read a vignette which described multiple stalking behaviours perpetrated by an individual against a former partner (described below). A between-groups design was used to manipulate the stalker and victim’s genders and sexuality. Participants were randomly allocated to one of four conditions: male perpetrator stalking female ex-partner (MSF, *n* = 86), female perpetrator stalking male ex-partner (FSM, *n* = 83), male perpetrator stalking male ex-partner (MSM, *n* = 81), and female perpetrator stalking female ex-partner (FSF, *n* = 86).

### 2.2. Materials

The experiment comprised of a vignette, six scale items measuring participants’ perceptions of the stalking behaviours, demographic questions (age, gender, and sexual orientation), and the stalking myth acceptance scale.

Stalking myth acceptance: The stalking myth acceptance scale ([18]) captures individuals’ attitudes towards ex-partner stalking through three dimensions of stalking myths. Participants rate their agreements with 22 statements on a 7-point scale (1 = strongly disagree to 7 = strongly agree). The *victim blaming* dimension reflects the extent to which individuals believe that victims of stalking are responsible for the stalker’s actions (e.g., “A woman who dates a lot would be more likely to be ‘stalked’”). The *flattery* dimension measures beliefs that stalking is a positive reaction towards women (e.g., “Women find it flattering to be persistently pursued”). The *minimising stalking* dimension captures the extent to which individuals downplay the harmfulness of stalking (e.g., “Stalkers are a nuisance, but they are not criminals”). Scores for each respective dimension’s items were averaged to calculate the final scores, with all three dimensions’ Cronbach α score demonstrating good internal reliability (victim blaming = 0.722; flattery = 0.801; stalking minimization= 0.974).

Vignettes: This study adopted a vignette used in [42] ([42]). The scenario described multiple stalking actions carried out by a perpetrator against a former partner, including repeated phone calls/emails, following, contacting the recipient’s friends, and approaching them in public. With the exception of the stalker and victim’s genders, the rest of the vignette details remained identical across all four conditions. Depending on the condition, the stalker was either presented as Katie, a female (FSF and FSM), or Steve, a male (MSF and MSM), and the recipient was presented as either Jenny, a female (MSF and FSF), or Peter, a male (FSM and MSM). An example of a vignette (MSF) is provided below:

“Jenny had been in a serious relationship with her boyfriend, Steve, for about 18 months before she decided to break up with him. She realised that they wanted different things from the relationship. Since breaking up with Steve two months ago, Jenny has received 20 or so calls and e-mails at work in which Steve asks her to take him back. Jenny also discovered that Steve had contacted her friends to see whether she mentioned him in conversations. There have been a few occasions when Steve has got on the same bus as Jenny in the morning and although he does not ask to sit next to her, he always makes eye contact and sits close by. Most recently, Steve approached Jenny while she was walking a friend’s dog in the local park and asked her to change her mind even though Jenny had made it clear that she is not interested”.

Attitudes towards vignettes: Participants’ perceptions of the vignette incidents were measured through six scale items that were measured on an 11-point scale. Eleven points allowed this study to measure attitudinal differences on a broader spectrum and the middle point (6) allowed participants to indicate uncertainty. Five of these items were adopted from [42] ([42]), measuring participants’ perceptions towards whether the behaviours constituted stalking (*Stalking*; 1 = “definitely not stalking” to 11 = “definitely stalking”), necessitated police intervention (*Intervention*; 1 = “definitely not” to 11 = “definitely”), would cause fear or apprehension (*Fear*; 1 = “not at all likely” to 11 = “extremely likely”), would cause mental or physical harm (*Harm*; 1 = “not at all likely” to 11 = “extremely likely”), and whether they were encouraged by the recipient (*Responsibility*; 1 = “not at all responsible” to 11 = “extremely responsible”). The five items, whilst covering different important elements of stalking perceptions, failed to consider how individuals would perceive possible risks of behaviour escalation. Thus, an additional question was added which asked participants to indicate whether they felt that the perpetrator’s behaviours were likely to escalate to a point where the recipient’s safety would be in danger (*Risk*, 1 = “definitely not” to 11 = “definitely”).

### 2.3. Procedure

After providing informed consent, participants were first asked to complete the stalking myth acceptance scale. Afterwards, they were randomly presented one of the four vignettes. Participants were instructed to take their time reading the vignette and make sure they had read and understood the incident carefully. Participants were then asked to answer the six vignette items. Finally, participants were required to complete a manipulation check question that asked them to confirm what they believed the perpetrator/recipient gender dynamic was. All participants correctly passed the manipulation check. Finally, participants were asked to answer a list of demographic questions (e.g., age, gender, and sexual orientation), after which this study was concluded and participants were debriefed.

## 3. Results

### 3.1. Stalker/Victim Relationship Dynamic

Average responses for all scale items by experimental condition are presented in Table 1. Although the distribution of all outcome variables deviated from normality, mean item ratings are presented alongside the median values to illustrate differences between experimental conditions. A series of Kruskal–Wallis tests were conducted to determine whether participants’ interpretations of the staking behaviour was affected by the perpetrator/recipient relationship dynamic. Significant group differences were only observed for *Intervention* (*p* = 0.003) and *Fear* (*p* < 0.001) responses (see Table 1). Post hoc Mann–Whitney tests were carried out to determine which conditions differed significantly; inferential results along with mean ranks are presented in Table 2.

Intervention responses: Mann–Whitney tests indicated that the need for police intervention was more strongly recommended in the MSF condition compared to the FSM condition (*p* = 0.002, *r* = −0.24), MSM condition (*p* = 0.003, *r* = −0.23), and FSF condition (*p* = 0.004, *r* = −0.22). No significant differences in *Intervention* responses were observed when the FSM, MSM, and FSF conditions were compared against each other.

Fear responses: Mann–Whitney tests indicated that participants believed that the perpetrator’s behaviours would create fear and apprehension to a greater extent in the MSF condition compared to the FSM condition (*p* < 0.001, *r* = −0.35), the MSM condition (*p* = 0.02, *r* = −0.18), and the FSF condition (*p* = 0.032, *r* = −0.17). The FSM condition produced the lowest responses, with participants in this condition also responding significantly lower on the *Fear* item than participants in the MSM condition (*p* = 0.006, *r* = −0.21) and FSF condition (*p* = 0.008, U = 2728.5, *r* = −0.21). No differences in *Fear* responses were observed between the MSM and FSF conditions.

### 3.2. Participant Gender and Sexuality

Gender differences in vignette responses were tested using a series of Mann–Whitney tests. Due to a low cell count, non-binary participants were not included in the gender comparison analyses (reduced *n* = 331). Significant gender differences were only observed for Stalking responses (Z = −2.59, *p* = 0.01, U = 6704, *r* = −0.2), with male participants (Mean Rank_Stalking_ = 193.59) agreeing more strongly with defining the perpetrator’s actions as stalking than female participants (Mean Rank_Stalking_ = 159.51).

Sexual orientation differences in vignette responses were tested through a series of Mann–Whitney tests. Due to low group sizes, sexual orientation was dichotomised into heterosexual (83.3%, *n* = 280) and groups with minority sexual orientation (14%, *n* = 47). A significant effect for participant sexual orientation was found for *Stalking* (Z = −3.08, *p* = 0.002, U =4767, *r* = −0.17) and *Risk* items (Z = −2.3, *p* = 0.021, U = 5212.5, *r* = −0.13). Participants reporting minority sexual orientation were more likely to define the perpetrator’s behaviours as stalking (Mean rank*_Stalking_* = 202.57) and more likely to see a risk of escalation (Mean rank*_Risk_* = 193.1) than heterosexual participants (Mean rank*_Stalking_* = 157.53; Mean rank*_Risk_* = 159.12).

### 3.3. SMA Predictors of Vignette Responses

A series of multiple linear regression models were tested to determine how well the SMA constructs (victim blaming, flattery, and stalking minimisation) could predict responses towards the stalking scenario for each scenario (MSF, FSM, MSM, FSF). Preliminary analyses were conducted to ensure no violation of the assumptions of linearity and homoscedasticity. The collinearity statistics (VIF and Tolerance) for all models indicated that multicollinearity was unlikely to be a problem (Tolerance > 0.1 and VIF > 10 for all predictors; see [52]). The SMA constructs successfully predicted all item responses within the MSF scenario, but only certain item responses within the other scenarios (see Table 3). The significant predictors for all responses are explained below. For conciseness, only the significant predictors of outcome variables are presented below.

In regard to labelling the behaviour as stalking, the SMA constructs were only able to predict responses for the MSF scenario. In this model, participants scoring high on the stalking minimisation construct were less likely to define the scenario as stalking (*β* = −0.34, *p* = 0.015), and participants scoring high on the victim blaming construct were more likely to label the scenario as stalking *(β* = 0.3, *p* = 0.046).

Taking notice of the need for police intervention, the SMA constructs were able to significantly predict responses in the MSF and FSF conditions. In the MSF scenario, participants scoring high on the stalking minimisation construct were less likely to agree that police intervention was necessary (*β* = −0.33, *p* = 0.014). Within the FSF condition, a greater belief in the flattery construct predicted disagreement with the need for police intervention (*β* = −0.43, *p* = 0.015), whilst greater belief in the victim blaming construct predicted greater agreement with the need for police intervention (*β* = 0.29, *p* = 0.048).

Perceptions of fear and apprehension were significantly predicted by the SMA constructs within the MSF scenario only, whereas greater belief in stalking minimisation was associated with greater disagreement that the victim (within the scenario) was likely to feel fear or apprehension as a result of the stalking (*β* = −0.46, *p* < 0.001). A similar predictive relationship was found for perceptions of harm and risk of escalation, whereby the SMA constructs were able to predict participant responses within the MSF scenario only. More specifically, participants holding greater belief in stalking minimisation reported greater disagreement that the victim was harmed (*β* = −0.33 *p* = 0.013) or that there was a significant risk of escalation (*β* = −0.39, *p* = 0.006).

Participants’ beliefs about whether the victim was responsible for the stalker’s actions (victim blaming) was significantly predicted by the SMA constructs within the MSF, MSM, and FSF conditions. For scenarios involving a female victim (MSF and FSF), participants scoring high on the victim blaming construct where more likely to assign responsibility to the victim for the event. This effect was greater within the MSF scenario (*β* = 0.6, *p* < 0.001) compared to the FSF scenario (*β* = 0.4, *p* = 0.005). Within the MSM scenario, participants with greater belief in the stalking minimisation construct were more likely to assign responsibility to the male victim (*β* = 0.38, *p* = 0.015).

## 4. Discussion

### 4.1. Effects of Stalker and Victim Gender on Participant Perceptions

The authors predicted that participants would identify stalking scenarios involving a male perpetrator (Hypothesis 1) or a female victim (Hypothesis 2) as the most pervasive (i.e., inducing fear, causing harm, and requiring police intervention). The results demonstrated a clear impact of the perpetrator and victim’s gender on stalking perceptions; however, this was only in relation to perceived fear and the need for police intervention. Male-stalking-female scenarios were rated as the most likely to induce fear in the victim and most likely to require police intervention. Furthermore, scenarios where a male stalked another male were seen as more fear-inducing in contrast to a female stalking a male. Likewise, when the stalker was presented as a female, perceptions of victim fear were far less for male victims compared to female victims.

Females who were stalked by females were perceived to be more fearful than males stalked by females. However, gender differences in the perceived need for police intervention were only observed for cases involving a male stalking a female. Thus, the collective findings partially support the first and second hypotheses.

Observations made in the present study correspond with previous research (e.g., [6]; [45]). [45] ([45]) also found that the perceived necessity of police intervention and perceived fear were greatest for cases of male-perpetrated stalking, whilst female-perpetrated stalking induced less fear. However, the findings suggest that male-perpetrated stalking was perceived as more likely to generate distress and fear of harm for the victims, whereas the equivalent measure from the present study (belief that the stalker caused physical or mental harm) was not influenced by the gender of stalker or victim. These differences could be due to the present study measuring perceptions of “physical or mental harm”, whereas [45] ([45]) measured perceptions of “distress and being alarmed”. The choice of wording used in the present study (mental harm) could convey a more detrimental impact which participants from the present study did not consider to be influenced by the genders of those involved. The authors acknowledge a further limitation with the decision to combine physical and mental harm in one measurement because the vignette did not describe any explicit physical altercations, and this may have conditioned the overall response.

It is possible that there is a common consensus that female-perpetrated stalking is perceived as non-threatening ([6]). However, this is contradicted by stalking perpetration data, which has shown female-initiated stalking has been reported to demonstrate similar, if not greater, risk patterns and risks of violent behaviour ([40]; [53]). It has also been proposed that males have an increased physical capacity to inflict and tolerate harm compared to females ([37]). This could explain the increased blame attribution towards male perpetrators and perceived decreased risk of harm for male victims *of stalking* perpetrated by females in cases of intimate partner stalking. It also highlights the importance of the potential influence of perpetrator and victim gender and/or other aspects of gender role beliefs on the perceptions and mitigation of risk, fear-escalating factors, perceived necessity of police intervention, and justification or acceptance of stalking behaviours between former intimate partners.

The general consensus for stalking definition involves some presence of fear and perceived implicit risk of physical harm ([58]). The findings of the present study suggest that individuals perceive male victims of intimate-partner stalking (especially FSM) to be less fearful than females who are stalked by their ex-partners. This highlights further implications for exploring a more robust consensus on a more objective and accurate stalking definition and whether perceived victim fear is subjective and therefore not a reliable factor. It also is important to take into consideration whether adverse victim emotional responses could be accounted for by individual factors.

### 4.2. Gender and Sexual Orientation Differences in the Perceiver

The third hypothesis predicted female participants would report attitudes that were more supportive towards stalking victims. The finding of the present study did not support this hypothesis. The only observable gender difference in responses suggested that males, when compared to females, were more likely to identify a stalker’s behaviour as stalking. This finding contradicts prior research, which has typically shown female participants to be more willing to identify stalking behaviours as stalking ([15]; [22]; [45]). However, much of the contradictory past research is relatively dated; thus, the conflicting present findings could signify major social changes in dating/relationship attitudes—though further corroborating evidence in support of this explanation is needed. Also, given that males are more frequent perpetrators of stalking ([3]), one possible explanation for our findings is that male participants may have been more familiar with behavioural indicators of stalking (i.e., through personal experience or awareness of others who had engaged in stalking). Whilst this observation is significant and of interest, the authors acknowledge that the observable effect size of this difference was small, suggesting negligible real-world differences.

In relation to sexual orientation, participants from sexual minority groups were more likely to identify the stalker’s behaviours as stalking and identified a greater risk of stalker’s behaviours escalating into violent behaviour. Though there is limited research looking into sexuality differences in stalking attitudes, findings from [48] ([48]) suggest that individuals from sexual minority groups were more likely to identify certain intrusive behaviours as being problematic. Research does suggest that individuals with minority sexual orientations are more likely to experience stalking ([12]; [19]; [48]). Thus, it is possible these individuals are more aware of stalking and how it escalates and are therefore more likely to identify early, non-violent signs of stalking as stalking behaviours and perceive a risk of further escalation. This aligns with defensive attribution theory, which dictates that individuals are more likely to identify with victims of crimes when they perceive equal levels of risk. However, women are also at greater risk of stalking than men, yet the aforementioned findings did not find similar differences between the gender groups. These findings were based on an intimate partner vignette; therefore, it is not clear if sexual minority individuals show a greater readiness to identify stalking behaviours (compared to heterosexual individuals) for all forms of stalking or only in cases involving ex-partners. Future research replications using additional stalking dynamics would help provide further insight into understanding the extent to which and reasons why sexual minorities are more likely to identify the behaviour of stalkers as stalking.

### 4.3. Application of the SMA to Non-Stereotypical Stalking

The SMA subscales were successful at predicting participant responses to all perception questions; however, this was only found within the scenario involving a male stalking a female. The subscales were able to report some responses within the other conditions, but this was inconsistent and only observed for a small proportion of the items. This demonstrates that intimate partner stalking myths will be influenced by beliefs and attitudes towards different gender groups rather than the crime itself. In other words, gender is the salient driver for these problematic beliefs, rather than someone’s general views on stalking or courting. [18] ([18]) reported that gender stereotype endorsement was associated with intimate partner SMA. It is certainly possible that gender views, namely sexism, may be a mediating factor for stalking myth acceptance. This has also been observed within the context of other gender-based crimes, with studies indicating that hostile sexism is correlated with myths surrounding both sex trafficking ([33]) and rape ([34]), with the mentioned studies finding that hostile sexism was a key factor in explaining individual differences in sex trafficking myths. Unfortunately, the present study did not measure sexism; nevertheless, it is an interesting direction for future research. It is possible that stalking myths regarding stalking involving other gender/sexual orientations may also exist; however, the current SMA scales cannot be used to measure them. As such, future research should attempt to develop a more inclusive measure of intimate partner SMA that covers other stalking genders and sexuality scenarios.

Looking more specifically at the predictors, it appeared that stalking minimisation was the most consistent predictor of problematic stalking attitudes. This is expected as the perception items are all related to perceptions of severity. Past research has shown that greater exposure to and awareness of stalking can reduce tendencies to minimise the severity of stalking ([44]; [57]). Together, these findings highlight that educational interventions that expose individuals to the reality of intimate partner stalking incidents (as well as other forms, e.g., stranger stalking) could serve to improve public attitudes towards stalking. This is especially important for individuals who may encounter victims or perpetrators of stalking through their occupations (e.g., police and social workers).

Participants who demonstrated greater victim blaming tendencies were more likely to identify the stalking ex-partner’s behaviours as stalking within the MSF condition. Though this shows a willingness to identify stalking offences, it does not imply that the individuals were more likely to hold supportive attitudes towards victims (i.e., individuals with victim blaming tendencies acknowledge that stalking is happening but believe that it is happening as a result of the victim’s actions). The victim blaming subscale from the SMA scale predicted participants’ decisions to assign blame on the victim in both the MSF and FSF conditions, but not the male victim scenarios. This suggests that participants endorsing female victim blaming beliefs may perceive female victims to be at fault for encouraging stalking behaviours from ex-partners. This corresponds with prior research, which suggests that victim blaming stems from hostile sexism towards women (i.e., perceiving women as problematic, see [33]).

### 4.4. Limitations and Directions for Future Research

The findings highlight the need for a stalking myth measurement tool that considers attitudes towards non-stereotypical stalking. The design of such a tool may enable a more holistic and contemporary understanding of stalking myths and beliefs.

The impact of age is an important factor that is frequently considered in legal and criminological research examining behaviours and attitudes relating to crime (e.g., [32]). The current study sample was composed of primarily young adult participants (*M* = 28.62, *SD* = 11.5), with the majority (*n* = 193) being active university students. Recent research by [36] ([36]) indicated that middle-aged adults hold greater stalking acceptance attitudes in comparison to younger adults. Furthermore, many universities are proactively attempting to reduce harassment and sexual misconduct on campuses; within the UK, the Office for Students have issued new requirements for universities to disclose the strategies being used to tackle such behaviours ([35]). As such, it is likely that student participants may also have had greater awareness about stalking behaviours. Research has also identified wider age differences in gendered attitudes and expectations which can inform perceptions of crime victims (see [25]). Together, this body of evidence suggests that younger adults, namely students, may reflect different attitudes to older adults and as such, the current sample may not fully represent the general population. As such, it is important for future research to consider how an individual’s age may influence their assessment of stalking severity.

There were methodological setbacks to the present study. Participant gender and sexuality groups were disproportionately recruited, with the majority of participants being female and heterosexual. Resultantly, the authors were unable to capture the attitudes of sexual minority groups such as non-heterosexual males. This further limited the ability to gain information pertaining to myths and beliefs surrounding stalking severity. Finally, the written vignette may have impacted the potential results attainable. Written vignettes are an effective method for gathering accurate insight into how individuals may respond to specific crimes (e.g., [16]; [54]). Nevertheless, the vignette used in the present study was short and may not have had a significantly immersive effect. As such, future research should adopt more contemporary approaches to presenting criminal scenarios, such as the use of videos depicting or describing criminal incidents.

## 5. Conclusions

Gender differences in perceptions of stalking severity appeared to be influenced by the gender of the perpetrator and victim, whereby stereotypical stalking was reported to induce greater levels of fear into victims and may result in a greater need of police intervention. The implications of such findings suggest that the perception of fear that is used in the differing definitions of stalking, alongside a lack of stalking interventions may place possible victims at risk of harm. These differential views may be a result of societal attitudes surrounding gender roles; however, future research is needed to ascertain this link. It is pivotal for researchers and victim advocates to continue working towards ensuring that stalking, and more specifically the needs of victims, is taken seriously regardless of the gender or sexuality of those involved. Thus, future research in this area is of clear value.

## Figures and Tables

**Table 1 behavsci-15-00120-t001:** Descriptive and inferential (Kruskal–Wallis) data for item responses between conditions.

	Stalking	Intervention	Fear	Harm	Responsibility	Risk
**MSF**						
Mean (SD)	8.62 (2.6)	6.88 (2.91)	8.71 (2.65)	7.69 (2.72)	1.92 (1.62)	8.12 (2.8)
Median (IQR)	9 (4)	7 (3)	9.5 (3)	8 (4)	1 (1)	9 (4)
**FSM**						
Mean (SD)	8.08 (2.14)	5.59 (2.69)	7.22 (2.44)	6.92 (2.18)	1.99 (1.74)	7.4 (2.31)
Median	8 (2)	6 (5)	8 (4)	7 (2)	1 (2)	7 (3)
**MSM**						
Mean (SD)	8.36 (2)	5.69 (2.47)	8.27 (2.02)	7.42 (2.38)	1.99 (1.36)	7.75 (2.2)
Median	8 (2)	6 (4)	8 (3)	8 (3)	1 (2)	8 (3)
**FSF**						
Mean (SD)	8.58 (2.07)	5.73 (2.69)	8.24 (2.18)	7.42 (2.34)	2.22 (1.78)	7.53 (2.56)
Median	9 (3)	6 (4)	8 (3)	8 (3)	1 (2)	8 (3)
Main effect (H,df = 3)	5.96	13.61 **	22.56 ***	6.14	2.55	6.54

Note. ** < 0.01, *** < 0.001.

**Table 2 behavsci-15-00120-t002:** Post hoc pairwise comparisons (Mann–Whitney-U).

Comparison	Intervention		Fear	
	Statistics	Mean Ranks	Statistics	Mean Ranks
MSF/FSM	Z = −3.09 **, U = 2594	96.34 vs. 32.25	Z = −4.52 ***, U = 2148.5	101.52 vs. 67.89
MSF/MSM	Z = −2.98 **, U = 2559	94.74 vs. 72.59	Z = −2.33 *, U = 2765	92.35 vs. 75.14
MSF/FSF	Z = −2.89 **, U = 2759	97.42 vs. 75.58	Z = −2.15 *, U = 3007.5	94.53 vs. 78.47
FSM/MSM	Z = −0.37, U = 3250.5	81.16 vs. 83.87	Z = −2.73 **, U = 2541	72.61 vs. 92.63
FSM/FSF	Z = −0.28, U = 3479.5	83.92 vs. 86.04	Z = −2.67 **, U = 2728.5	74.87 vs. 94.77
MSM/FSF	Z = −0.01, U = 3479.5	84.04 vs. 83.96	Z = −0.04, U = 3472	84.14 vs. 83.87

Note. * < 0.05, ** < 0.01, *** < 0.001.

**Table 3 behavsci-15-00120-t003:** Multiple linear regression models for vignette responses.

	MSF	FSM	MSM	FSF
Stalking	F = 4.43, R^2^ = 0.154 **	F = 0.53, R^2^ = 0.021	F = 0.511, R^2^ = 0.021	F = 2.09, R^2^ = 0.075
Police intervention	F = 7.24, R^2^ = 0.229 ***	F = 0.307, R^2^= 0.012	F = 0.19, R^2^ = 0.008	F = 4.02, R^2^ = 0.135 *
Fear	F = 9.72 R^2^ = 0.285 ***	F = 0.35, R^2^ = 0.014	F = 0.35, R^2^ = 0.014	F = 2.66, R^2^ = 0.094
Harm	F = 8.24, R^2^ = 0.253 ***	F = 1.2, R^2^ = 0.046	F = 2.34, R^2^ = 0.09	F = 0.79, R^2^ = 0.03
Victim blame	F = 14.75, R^2^ = 0.377 ***	F = 0.5, R^2^ = 0.02	F = 6.09, R^2^ = 0.21 ***	F = 7.65, R^2^ = 0.23 ***
Escalation	F = 4.67, R^2^ = 0.16 **	F = 0.1, R^2^ = 0.004	F = 0.69, R^2^ = 0.028	F = 2.09, R^2^ = 0.075

Note. *Df* = 3, *** = *p* < 0.001, ** = *p* < 0.01, * = *p* < 0.05.

## Data Availability

Data are available upon request from the corresponding author.

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
