# Peer review of "Impact of Perpetrator and Victim Gender on Perceptions of Stalking Severity"

_behavsci, 2025, doi:10.3390/bs15020120_

Round 1

Reviewer 1 Report

Comments and Suggestions for Authors

Congratulations on a great piece of research which communicated with coherence and is meticulously structured.  Your focus on LGBTQ+ communities’ perceptions of stalking is commended, as you state there is a dearth of research here despite an increased vulnerability to stalking which has been widely acknowledged.

Some suggestions for you:

Ex partner stalking makes up about 50% of stalking cases, as you will know, the other forms of stalking are often overlooked by the public and responded to with less rigour by services that rely on a DA model of safeguarding, it would be helpful to acknowledge that and position ex-partner stalking in that wider context at the beginning of the paper, before focusing specifically on ex-partner stalking.

Limitations and directions

Your study did produce some unexpected results, and you discuss the role of sample size in this. But what other factors might have influenced this?

In line 468 you address age as an important area to investigated in terms of stalking perceptions, how might the age of your, mostly young respondents have influenced the perceptions that were recorded. It would also be helpful to see the restrictions of the sample discussed in more detail. Respondents were psychology graduates from a single institution, much work on stalking has been based on university groups due to their availability, but they are often presented as just that rather than as general a general group. E.g  Fremouw, W. J., Westrup, D., & Pennypacker, J. (1997). Stalking on campus: The prevalence and strategies for coping with stalking. Journal of Forensic Sciences42(4), 666-669.

The development of the limitations sections would help to mitigate this. You may wish to consider a change in the title as well to highlight your sample, Universities in the UK have conditions of registration set by the OFS that require them to safeguard students and stalking is an area where they should be up to speed, this paper has considerable value in that space.

Author Response

Comment 1: Ex partner stalking makes up about 50% of stalking cases, as you will know, the other forms of stalking are often overlooked by the public and responded to with less rigour by services that rely on a DA model of safeguarding, it would be helpful to acknowledge that and position ex-partner stalking in that wider context at the beginning of the paper, before focusing specifically on ex-partner stalking.

Response 1: we have provided further content explaining the other two types of stalking. However, evidence shows that both police and lay persons view stranger stalking as more pervasive and serious, whilst also being more dismissive towards ex-partner stalking. We provide evidence for this in the introduction.

Comment 2: Your study did produce some unexpected results, and you discuss the role of sample size in this. But what other factors might have influenced this?

Response 2: The authors have offered further explanations for the gender differences in stalking identification.

Comment 3: In line 468 you address age as an important area to investigated in terms of stalking perceptions, how might the age of your, mostly young respondents have influenced the perceptions that were recorded. It would also be helpful to see the restrictions of the sample discussed in more detail. Respondents were psychology graduates from a single institution, much work on stalking has been based on university groups due to their availability, but they are often presented as just that rather than as general a general group. E.g  Fremouw, W. J., Westrup, D., & Pennypacker, J. (1997). Stalking on campus: The prevalence and strategies for coping with stalking. Journal of Forensic Sciences, 42(4), 666-669. The development of the limitations sections would help to mitigate this. You may wish to consider a change in the title as well to highlight your sample, Universities in the UK have conditions of registration set by the OFS that require them to safeguard students and stalking is an area where they should be up to speed, this paper has considerable value in that space.

Response 3: Thank you for the information regarding the OFS requirements, this was very useful. We have provided more context here to explain how and why the skewed age group will  have impacted the generalisability of the findings. We acknowledge that a large proportion of the sample were students but many of others were not students as well (we provide the statistics in the methodology now). Thus we don’t think the title of the paper should be changed.  However, we do acknowledge the effects of student samples in the limitations.

Reviewer 2 Report

Comments and Suggestions for Authors

Overall I thought this well a well written and engaging article that contributes to the literature with well presented findings. My recommendation is that minor amendments are made, particularly to the introduction. I’ve included specific details below, but overall I would suggest that the emphasis on ex-partner stalking needs to be strengthened throughout. This includes adding the fact your work focuses on ex-partner stalking to the title, and flagging this more throughout the abstract (you also don’t have anything about participant gender or sexuality in the title – may be trickier to include but worth considering). There is also no discussion of the fact you’ve focused on ex-partner stalking in the discussion and how this may have affected the results. E.g. the relationship between rape mythology, the real rape stereotype, and the idea that only stranger rapes are ‘real’ rapes, for instance, may be relevant here in picking apart how the presentation of an ex-partner vignette may have influenced the results. 

INTRODUCTION

Bottom of page 1 / start of page 2 – more likely to accept behaviour as stalking if it occurs shortly after the relationship ends – compared to some time after the relationship has ended? Some clarity needed here as it’s between two sentences which state former partner stalking is not perceived as such.

As the focus is within the context of intimate partner relationships, can more clear links be made between this relationship type and the myth constructs? Making more of the focus on ex-partner stalking here would be helpful. You note Dunlap’s scale is specific to measure ex-partner stalking specifically, but only on the second mention on P2 L87, and not when first introduced higher up. Introducing this fact earlier would help with the reason for why myth scales have been developed specifically for different offender-victim relationship types, which it would be helpful to more explicitly outline.

Section 1.1 – with the focus on ex-partner stalking, there is an argument for bringing in the literature here on domestic offending more generally and how seriously (or not) domestic abuse is taken when it is a female perpetrator and male victim.

Hypothesis 1 – you note in the introduction that male stalking behaviour is more likely to be normalised, so what is the reason for hypothesising that male stalking will be taken more seriously? Likewise with Hypothesis 2 – male victim behaviour may be downplayed but female victims are more often blamed (from introduction) so what is the reason for the direction of this hypothesis? The use of the word ‘problematic’ here is also vague and needs to relate more clearly to the measurements used in the Method (the word problematic is used in the discussion too and is also vague here – considering replacing). Severity could be better, as it’s used in the title?

METHOD

Really clearly and concisely written.

RESULTS

A little information on the assumptions testing for the regression analyses would be useful, but again, this section is really clearly written.

DISCUSSION

Briefly revisiting what the hypotheses are when you talk about whether they’ve been met or not would be helpful for the reader.

I find the explanation regarding hypothesis 3 findings, i.e. the effect size was small, to be a little weak considering you found a significant difference. You don’t necessarily need to add in more reasoning for your findings, but at the least I would remove this explanation.

You’ve discussed sexism and adherence to myths; some reference to the rape myth literature as well as the human trafficking literature here would be useful. Ditto in the section on victim blame – there’s a lot of rape myth literature to draw on here.

As noted above, some reference to your focus on ex-partner stalking is required in this section. In places this can be as simple as e.g. saying ‘The findings highlight the need for an ex-partner stalking myths measurement’. Some expansion on how the presentation of a certain relationship type is also needed here.

Some small formatting points:

-          P2 L65 – themselves, not their selves

-          P3 L114 – after normalised and before Whereas there should be a comma, not a full stop, again before Whereby on P3 L137, again before Whereas on P8 L337, and again before Whereas on P9 L371 (these are currently not complete sentences).

-          Some small reformatting of in text citations required, e.g. you don’t need a comma after et al. when it’s in the main text, you don’t need a full stop after a name, you do need both a full stop and a comma after et al when in brackets (if you’re following APA formatting – you may be using a different formatting but the in text citation formatting needs to be consistent).

-          P8 L320 – predicted, not predict

-          P9 L360 – fear in the victim, not fear onto the victim

-          P9 L362 – ‘Likewise, females who were stalked by females were perceived to be more fearful than males stalked females.’ This sentence is a little unclear.

-          P9 L385 – should read ‘perceived decreased risk of harm for male victims of stalking perpetrated by females.

-          P10 L409 – remove the word ‘with’ (currently reads ‘participants with from’)

-          P10 L426 – The subscales, not There subscales

-          P11 L481 – vignettes, not vignette (or vignette and was, not were)

Author Response

Comment 1: Bottom of page 1 / start of page 2 – more likely to accept behaviour as stalking if it occurs shortly after the relationship ends – compared to some time after the relationship has ended? Some clarity needed here as it’s between two sentences which state former partner stalking is not perceived as such.

Response 2: What the authors were trying to communicate was that participants have been found to excuse ex-partner stalking behaviours that occur straight after a relationship ends due to the behaviour being seen as an attempt to save a relationship (rather than malicious stalking). This section has been revised to enhance clarity.

Comment 2: As the focus is within the context of intimate partner relationships, can more clear links be made between this relationship type and the myth constructs? Making more of the focus on ex-partner stalking here would be helpful. You note Dunlap’s scale is specific to measure ex-partner stalking specifically, but only on the second mention on P2 L87, and not when first introduced higher up. Introducing this fact earlier would help with the reason for why myth scales have been developed specifically for different offender-victim relationship types, which it would be helpful to more explicitly outline.

Response 2: We have now made it clearer that intimate partner stalking can invoke unique stalking myths. This is explained earlier on. Further explanation of how the three constructs relate to intimate partner stalking are also provided. Due to a lack of qualitative evidence around specific intimate partner stalking , the authors were cautious not to turn these sections into speculative assertions, therefore, the additional content is limited but serves to address the issue raised.

Comment 3: Section 1.1 – with the focus on ex-partner stalking, there is an argument for bringing in the literature here on domestic offending more generally and how seriously (or not) domestic abuse is taken when it is a female perpetrator and male victim.

Response 3: We have now provided some references to studies exploring attitudes towards female perpetrated IPV.

Comment 4: Hypothesis 1 – you note in the introduction that male stalking behaviour is more likely to be normalised, so what is the reason for hypothesising that male stalking will be taken more seriously?

Likewise with Hypothesis 2 – male victim behaviour may be downplayed but female victims are more often blamed (from introduction) so what is the reason for the direction of this hypothesis?

The use of the word ‘problematic’ here is also vague and needs to relate more clearly to the measurements used in the Method (the word problematic is used in the discussion too and is also vague here – considering replacing). Severity could be better, as it’s used in the title?

Response 4: We understand the confusion caused by the lack of clarity in our writing. What we were trying to convey was that the one qualitative study (Gavin & Scott, 2016) found a theme of beliefs which normalised male perpetrated stalking and another theme that presented female perpetrators (but not male perpetrators) as being more pathological. However, despite this finding, the overwhelming literature demonstrates that male perpetrated stalking is still seen as being more dangerous and serious. It is for this reason that we made our first hypothesis.

For the second hypothesis you are right in that Gavin and Scott (2016) found victim blaming to be more prevalent in female victimised scenarios, however, again, the overarching majority of studies (e.g., Cass & Mallicoat, 2015; Gavin & Scott, 2016; Sheridan, North & Scott, 2014; Strand & McEwan, 2011) suggest that female victimised scenarios are still seen as being more problematic. I think the main issue, as you indicate , is the poor choice of wording we used with ‘problematic’. This was an umbrella term reflecting beliefs of the behaviour being more harmful and fear inducing. We have now changed our choice of wording.

Comment 5: A little information on the assumptions testing for the regression analyses would be useful, but again, this section is really clearly written.

Response 5: Thank you, we have provided further information regarding the regression assumptions.

Comment 6: Briefly revisiting what the hypotheses are when you talk about whether they’ve been met or not would be helpful for the reader.

Response 6: The hypotheses are now reported at the start of the discussion section (hypothesis 3 is presented later).

Comment 7: I find the explanation regarding hypothesis 3 findings, i.e. the effect size was small, to be a little weak considering you found a significant difference. You don’t necessarily need to add in more reasoning for your findings, but at the least I would remove this explanation.

Response 7: We agree, the minimal effect size is not a sufficient explanation for the observation. We included this comment as a caveat rather than a concrete explanation. We have now provided an additional possible explanation for our observation.

Comment 8: You’ve discussed sexism and adherence to myths; some reference to the rape myth literature as well as the human trafficking literature here would be useful. Ditto in the section on victim blame – there’s a lot of rape myth literature to draw on here.

Response 8: The manuscript had included a few different sources that studied rape myths and sex trafficking myths (e.g., Mojtahedi et al., 2024). We have tried to limit how much we refer to this literature to 1)  keep the word count of the overall manuscript from becoming too long and 2)   because we felt that these points were sufficiently raised. With that said, we have added some additional citations to the discussion and Introduction. We hope that this is sufficient.

Comment 9: As noted above, some reference to your focus on ex-partner stalking is required in this section. In places this can be as simple as e.g. saying ‘The findings highlight the need for an ex-partner stalking myths measurement’. Some expansion on how the presentation of a certain relationship type is also needed here.

Response 9: As suggested, we have made more specific references to intimate-partner stalking within the discussion of our findings.

Comment 10: Formatting

     P2 L65 – themselves, not their selves

-          P3 L114 – after normalised and before Whereas there should be a comma, not a full stop, again before Whereby on P3 L137, again before Whereas on P8 L337, and again before Whereas on P9 L371 (these are currently not complete sentences).

-          Some small reformatting of in text citations required, e.g. you don’t need a comma after et al. when it’s in the main text, you don’t need a full stop after a name, you do need both a full stop and a comma after et al when in brackets (if you’re following APA formatting – you may be using a different formatting but the in text citation formatting needs to be consistent).

-          P8 L320 – predicted, not predict

-          P9 L360 – fear in the victim, not fear onto the victim

-          P9 L362 – ‘Likewise, females who were stalked by females were perceived to be more fearful than males stalked females.’ This sentence is a little unclear.

-          P9 L385 – should read ‘perceived decreased risk of harm for male victims of stalking perpetrated by females.

-          P10 L409 – remove the word ‘with’ (currently reads ‘participants with from’)

-          P10 L426 – The subscales, not There subscales

-          P11 L481 – vignettes, not vignette (or vignette and was, not were)

Response 10: We thank the reviewer for their diligent review. We have addressed all the above errors and proof-read the manuscript to make sure that the article is free from error.

Reviewer 3 Report

Comments and Suggestions for Authors

page 3: "The gender and sexual orientation of the observer (participant)" this is not very clear. Who exactly are you talking about?

In the methods section there are minor grammatical errors that need to be fixed. One example is on page 6: "Participants were instructed take their time reading the vignette and make sure 265 they had read and understood the incident carefully" need to include the work "to" before "take their time"

page 6 line 279: Significant should be Significance

page 6: "No significant differences in Intervention responses were observed between the FSM, MSM and FSM conditions." I am not sure which groups are being compared. FSM is in the sentence twice

Author Response

Comment 1: page 3: "The gender and sexual orientation of the observer (participant)" this is not very clear. Who exactly are you talking about?

Response 1: for clarity, we have now changed this sentence to “Perceptions of stalking incidents can also be influenced by the gender and sexual orientation of the perceiver”

Comment 2: In the methods section there are minor grammatical errors that need to be fixed. One example is on page 6: "Participants were instructed take their time reading the vignette and make sure 265 they had read and understood the incident carefully" need to include the work "to" before "take their time".

Response 2: This typo has now been amended and we have proof-read the manuscript to make sure that the article is free from errors.

Comment 3: page 6 line 279: Significant should be Significance

Response 3: Wording has now been changed so that the sentence makes sense.

Comment 4: page 6: "No significant differences in Intervention responses were observed between the FSM, MSM and FSM conditions." I am not sure which groups are being compared. FSM is in the sentence twice.

Response 4: This sentence has been revised and is now correctly presented.

Reviewer 4 Report

Comments and Suggestions for Authors

The article considered whether gender and sexuality of the perpetrator, victim and participant resulted in inaccurate/problematic stalking perceptions, also examining whether stereotypical stalking attitudes (through SMA scales) would predict stalking incidents' perceptions which involved female stalkers or male victims.

The article illuminates a relevant phenomenon and aims to contribute to further comprehension by adding the non-stereotypical victim-perpetrator of stalking. It is well-written, clear and concise; therefore, in my analysis, it should be accepted for publication, after some minor revisions. The major issue I find in this study is related to the use of non-parametric tests. I understand the use of non-parametric tests when the authors compare Male vs. Female or Heterossexual vs Sexual Minorities participants (since there is an evident N difference between these groups) However, even though the authors state that "Although the distribution of all outcome variables deviated from normality", I would rather see the differences among conditions to be analysed through parametric tests (namely ANOVAs). 

Furthermore, I believe "physical and emotional harm" should've not been measured together in a single item, but rather separately, especially considering the vignettes applied, where there is no evident physical contact between the victim-perpetrator, which might have conditioned the responses (putting myself in a participant role, I would consider the situation to be more emotionally stressful, but maybe not so much regarding physical harm). So this might be a limitation to be addressed. 

One of the most surprising results of the study was that "males were more likely when compared to females to identify a stalker’s behaviour as stalking.". However, this result is poorly addressed/analysed by the authors.  The more recent articles supporting otherwise are almost 10 years old. Considering the major social changes occurring all over, maybe there could other explanations for such an unexpected result.

Indeed, another relevant result was participants who demonstrated greater victim-blaming tendencies were more likely to identify the stalker’s behaviours as stalking within the MSF condition. Although this is not compared between participants' genders, being most of the participants female, which could provide some clues for the previously referred result.  

Last, the authors state that "age is an important factor that is frequently considered in legal and criminological research examining the performance and attitude of individuals on matters relating to crime", and that this was not analysed in the results. Why? The age range of the participants varies between 18-72. Maybe it would be interesting to perform some correlation analysis with age and the VDs, or to divide participants into age groups to run some comparisons (I don't know if that would be possible). 

Author Response

Comment 1: The major issue I find in this study is related to the use of non-parametric tests. I understand the use of non-parametric tests when the authors compare Male vs. Female or Heterossexual vs Sexual Minorities participants (since there is an evident N difference between these groups) However, even though the authors state that "Although the distribution of all outcome variables deviated from normality", I would rather see the differences among conditions to be analysed through parametric tests (namely ANOVAs).

Response 1: We understand that parametric testing would allow us to compare differences more intricately, however, given the skewness of our data, using such analyses would produce unreliable results. Whilst we appreciate that ANOVA’s can accommodate for slight variations from normality, we opted to use non-parametric analyses for more reliable results. This was also accepted by the three other reviewers so we hope that the current reviewer can accept our decision to stick with non-parametric testing.

Comment 2: Furthermore, I believe "physical and emotional harm" should've not been measured together in a single item, but rather separately, especially considering the vignettes applied, where there is no evident physical contact between the victim-perpetrator, which might have conditioned the responses (putting myself in a participant role, I would consider the situation to be more emotionally stressful, but maybe not so much regarding physical harm). So this might be a limitation to be addressed. 

Response 2: This is a valid critique that the authors have now acknowledged as a methodological limitation.

Comment 3: One of the most surprising results of the study was that "males were more likely when compared to females to identify a stalker’s behaviour as stalking.". However, this result is poorly addressed/analysed by the authors.  The more recent articles supporting otherwise are almost 10 years old. Considering the major social changes occurring all over, maybe there could other explanations for such an unexpected result.

Response 3: we have provided two further explanations for these findings including the valid suggestion made by the reviewer.

Comment 4: Indeed, another relevant result was participants who demonstrated greater victim-blaming tendencies were more likely to identify the stalker’s behaviours as stalking within the MSF condition. Although this is not compared between participants' genders, being most of the participants female, which could provide some clues for the previously referred result.  

Response 4: This is an interesting proposition, it is possible that the gender of the participants could have moderated the relationship between stalking myths and vignette responses. However, after careful consideration, the authors feel that proposing that the overrepresentation of female participants could have moderated this observation is heavily speculatory. We did consider including it as a direction for future research, but even so, there is not concrete rationale to support the need to consider the moderating effect of gender on the relationship between stalking myths and vignette perceptions. We feel that including such considerations would be too speculatory – though it is an interesting consideration for us to take on board in our future research.

Comment 5: Last, the authors state that "age is an important factor that is frequently considered in legal and criminological research examining the performance and attitude of individuals on matters relating to crime", and that this was not analysed in the results. Why? The age range of the participants varies between 18-72. Maybe it would be interesting to perform some correlation analysis with age and the VDs, or to divide participants into age groups to run some comparisons (I don't know if that would be possible).

Response 5: The section being referred to was presented as a direction for future research, rather than a limitation of the present study. There are two reasons why age was not analysed int the present study. The first is that it was not a central aim of the study and including it as an additional objective would shift the narrative from the overarching focus of the present study (i.e., gender and sexuality differences), not to mention that this would also require further consideration of relevant literature which would push the final word count well above the standard expected length. Secondly, despite having some participants that were elderly, the age spread of the sample was skewed towards younger individuals, as reflected by the mean and standard deviation, the majority of participants were aged 20-40. Thus, it would not be feasible to accurately examine for larger age group differences. The lack of variation in sample age is now acknowledged within our discussion.